# Mixed-Culture Propagation of Uterine-Tissue-Resident Macrophages and Their Expression Properties of Steroidogenic Molecules

**DOI:** 10.3390/biomedicines11030985

**Published:** 2023-03-22

**Authors:** Kazushige Ogawa, Takashi Tanida

**Affiliations:** Laboratory of Veterinary Anatomy, Graduate School of Veterinary Science, Osaka Metropolitan University, 1-58 Rinku-Ourai-Kita, Izumisano, Osaka 598-8531, Japan; t-tanida@omu.ac.jp

**Keywords:** uterine macrophage, SRD5A1, HSD17B1, StAR, HSD3B

## Abstract

Tissue-resident macrophages (Mø) play tissue/organ-specific roles, and the physiological/pathological implications of uterine Mø in fertility and infertility are not yet fully understood. Herein, we report a simple propagation method for tissue-resident Mø by mixed culture with the respective tissue/organ-residing cells as the niche. We successfully propagated mouse uterine Mø by mixed culture with fibroblastic cells that exhibited properties of endometrial stromal cells. Propagated mouse uterine Mø were CD206- and arginase-1-positive; iNOS- and MHC-II-negative, indicating M2 polarization; and highly phagocytic, similar to endometrial Mø. Furthermore, uterine Mø were observed to express steroidogenic molecules including SRD5A1 and exhibited gap junction formation, likely with endometrial stromal cells. Accordingly, uterine Mø propagated by mixed culture may provide a new tool for studying immune–endocrine interactions related to fertility and infertility, particularly androgen’s intracrine actions in preparing the uterine tissue environment to support implantation and pregnancy as well as in the etiology of endometriosis.

## 1. Introduction

Macrophages (Mø) are largely classified into the following two types in adults: tissue-resident Mø and recruited Mø [1]. Tissue-resident Mø, which originate from fetal precursors in the yolk sac and fetal liver (fetal Mø) as well as bone-marrow-derived blood monocytes (BMMs), colonize tissue/organ-specific microenvironments (the niche of residence for Mø) suitable for long existence and survival at steady state, and act tissue/organ-specifically to maintain tissue/organ homeostasis [2,3]. The niche provides nourishment, signal molecules, and cytokines, including colony-stimulating factor 1 (CSF1), CSF2, and/or interleukin-34 (IL34) to Mø, leading to their self-renewal and shaping them to possess tissue/organ-specific properties [2,3]. By contrast, recruited Mø originate from BMMs that infiltrate lesions in response to damage and inflammation in tissues/organs to resolve them [1].

Tissue-resident Mø colonize all three uterine layers (endometrium, myometrium, and perimetrium) and densely appear in the endometrium. In humans, leukocyte densities, including Mø, in the endometrium fluctuate during the natural menstrual cycle phases [4,5]. Estrogen and progesterone affect leukocyte populations, including Mø, in the endometrium [6]. Mø population fluctuations also occur in the rodent endometrium during the uterine cycle phases, although the menstrual phase is lacking in rodents [7,8]. Uterine Mø in adults are constantly replenished by BMMs [4,9]. To maintain tissue integrity during the natural menstrual cycle, Mø in the endometrium have been implicated in the regulated repair of the endometrium without scar formation, including the regeneration of the endometrial tissue and phagocytic clearance of cell/tissue debris in the endometrium [9,10]. Although clear evidence has not been accumulated on their uterine functions, endometrial Mø are likely involved in normal pregnancy in various processes such as trophoblast invasion and remodeling of the decidual tissues, including spiral arteries, as well as maintenance of immune tolerance [5,11]. Additionally, Mø are involved in the pathogenesis of endometriosis [12]: endogenous Mø of the M2 phenotype appear as players in the natural history of endometriosis, required for angiogenesis and ectopic lesion growth [13]; interactions between M2 Mø and endometrial stromal cells may play an important role in the development of endometriosis [14]. CSF1-deficient osteoporotic (op/op) mice and *Cd11b*-*tdr* transgenic mice have been used to reveal those roles by depleting uterine Mø in the mice [15,16,17,18]. However, the systemic depletion of Mø occurs in these mouse models, and moreover, the CSF1 deficiency likely affects the dendritic cell population because the CSF1 receptor (CSF1R) is expressed in the mononuclear phagocyte lineage [19], and the CD11b deficiency may affect the neutrophil population because CD11b is a common marker of myeloid cells [20]. Leucocytes regularly infiltrate into the uterus during the menstrual cycles/estrus cycles [4,9], and thus it may be difficult to refer to functions restricted to Mø colonizing the uterus of those models. Further accumulation of evidence is essential to clearly understand the precise and comprehensive roles of uterine Mø involved in those physiological and pathological events. In vitro studies using uterine Mø may be essential for the evaluation of uterine Mø in physiology and pathology. To the best of our knowledge, at present, sufficient uterine Mø are not available given the difficulty of their propagation in culture, and thus there are few in vitro studies examining the physiological and pathological implications of uterine Mø. 

Recently, we developed a propagation method of tissue-resident Mø by mixed culture with the respective tissue/organ-residing cells that could be applicable for several tissue-resident Mø [21] and successfully propagated lung interstitial Mø [22], testicular interstitial Mø [23], and Kupffer cells [24], all of which showed identical properties in marker expressions to those in respective ex vivo Mø. In these mixed cultures, tissue-resident Mø could propagate along with the propagation of organ-specific interstitial cells, which behaved as the niche for the respective Mø because these interstitial cells expressed *Csf1*, *Csf2*, and/or *Il34* as growth factors for Mø, as well as marker molecules specific for the respective niche cells. Therefore, organ-specific properties were likely maintained in those Mø propagated by mixed culture with the organ-specific interstitial cells. CSF1 not only stimulates the extravasation of monocytes but also induces the proliferation of Mø in the pregnant mouse uterus [25]. Moreover, CSF1 is produced by endometrial epithelial cells [26] and stromal cells [27]. Based on these findings on CSF1 expression in the uterus as well as those findings on the propagated Mø, we speculated that uterine-tissue-resident Mø could be propagated if a suitable niche producing CSF1 could be reproduced in vitro. Therefore, we tried to propagate uterine-tissue-resident Mø by mixed culture with uterine cells, including endometrial stromal cells (fibroblasts) and/or epithelial cells that may possibly provide the niche due to the expression of CSF1, applying our propagation method of tissue-resident Mø.

We recently demonstrated that testicular interstitial Mø propagated by mixed culture with fibroblastic cells, with properties of Leydig cells in culture, produce progesterone de novo [23] and found that the fibroblastic cells have an ability to shape other Mø to produce progesterone de novo (unpublished data). The uterus not only is a target organ of sex steroids, such as estrogen and progesterone, but also locally produces and converts/metabolizes sex steroids: sex-steroid-producing/converting/metabolizing molecules are expressed in endometrial epithelial and stromal cells [28]. Based on the evidence, we hypothesized that tissue-resident Mø colonizing around sex-steroid-producing cells such as endometrial stromal cells have the ability to produce sex steroids de novo and/or convert/metabolize sex steroids. In the present study, we evaluated this hypothesis by examining expressions of steroidogenic molecules in uterine Mø propagated by mixed culture with uterine fibroblastic cells, assumed to be the niche cells. Accordingly, we successfully demonstrated the expression of steroidogenic molecules in the uterine Mø.

## 2. Materials and Methods

### 2.1. Animals

Specific-pathogen-free ICR female mice were obtained from Japan SLC Inc. (Hamamatsu, Japan). We used 18 mice aged 8–9 weeks maintained under standard feeding and housing (12 h light/dark cycle; 22–23 °C) conditions. The animal experimentation protocol was approved by the Animal Research Committee of Osaka Metropolitan University (approval numbers: 21–26, 22–32), and all experiments were performed following the relevant guidelines of Osaka Metropolitan University. The phases of the estrous cycle (proestrus, estrus, metestrus, diestrus) were determined according to conventional vaginal smear/cytology [29] using Giemsa staining (Giemsa Stain Solution, FUJIFILM Wako Pure Chemical Corp., Osaka, Japan). After assessment, the mice were euthanized by cervical dislocation, and the uterus was aseptically dissected and immediately dipped into ice-cold Ca/Mg-free Hanks’ Balanced Salt Solution (HBSS; H6648, Sigma-Aldrich, St. Louis, MO, USA).

### 2.2. Propagation of Uterine Mø by Mixed Culture with Other Interstitial Cells from the Uterus

We propagated tissue-resident Mø from the mouse uterus according to our previously established method with some modifications [21]. Briefly, two-thirds of the uterus per mouse was minced into approximately 1 mm^3^ pieces, digested with 12.5 mL collagenase solution (0.5 mg/mL collagenase type IA [C9891, Sigma-Aldrich] and 1 mM CaCl_2_ in 20 mM HEPES-buffered HBSS, pH 7.35) at 37 °C for 40 min with gentle stirring, and then washed once with HBSS. Following further dispersion by pipetting, the cell/tissue suspensions were sedimented at 100× *g* for 5 min; seeded in a 10 cm tissue culture dish containing 12 mL DMEM (D6046, Sigma-Aldrich or 08456-65, Nacalai Tesque, Kyoto, Japan) supplemented with 10% fetal bovine serum (S1580-500, Biowest, Nuaillé, France), as well as 100 U/mL penicillin and 100 μg/mL streptomycin (P4333, Sigma-Aldrich) (FBS-DMEM); and cultured in a CO_2_ incubator. FBS-DMEM was refreshed every 2–3 days until the dish surface was occupied with multilayered cells composed mainly of fibroblastic cells and uterine Mø (usually within one week). Overconfluent cells were detached using 0.1% trypsin (T4674, Sigma-Aldrich) and 2 mM EDTA in 10 mM HEPES-buffered HBSS (Trypsin-EDTA), and the cells at a dilution ratio of 1:3 were sub-cultured with FBS-DMEM until they became overconfluent again (usually 7–14 days). In some cases, the cells were suspended in a cryopreservative (Bambanker; CS-02-001, Nippon Genetics, Tokyo, Japan) and frozen at −80 °C. 

### 2.3. Separation of Uterine Mø Propagated by Mixed Culture with Other Uterine Interstitial Cells

Uterine Mø propagated by mixed culture were segregated from other interstitial cells using our previously established method [21]. Briefly, the overconfluent cells in a 10 cm tissue culture dish were detached using Trypsin-EDTA and seeded in a 5.5 cm bacteriological Petri dish (1-8549-02, As One, Osaka, Japan) and cultured with FBS-DMEM. After several hours, when uterine Mø preferentially adhered to the dish surface and other uterine interstitial cells formed nonadherent cell aggregates in the dish, nonadherent cells/aggregates were removed by washing with conditioned media. The adherent cells were detached with 10 mM HEPES-buffered HBSS containing 5 mM EDTA, followed by pipetting. Subsequently, cells were passed through a cell strainer (352235, BD Falcon, Franklin Lakes, NJ, USA) to remove cell aggregates, centrifuged at 220× *g* for 5 min, and resuspended in HBSS (for phagocytosis analysis and reverse transcription-polymerase chain reaction [RT-PCR]) or in Ca/Mg-free phosphate-buffered saline (1102P10, Cell Science & Technology Institute, Yamagata, Japan) containing 1% bovine serum albumin (A3059, Sigma-Aldrich) and 2 mM EDTA (BSA-PBS; for flow cytometry). Then, the obtained cells were counted and used for further experiments.

### 2.4. Separation of Uterine Interstitial Cells Showing Fibroblastic Morphology from Uterine Mø

Uterine interstitial cells propagated by mixed culture with uterine Mø were segregated by the treatment with clodronate-encapsulating liposomes to deplete phagocytic cells using a method described by Yamauchi et al. [23] with minor modifications. Briefly, the mixed cultured cells that reached near-confluence were treated with 4 µL/mL clodronate-encapsulating liposomes (7 mg of clodronate/mL; Clophosome-A, F70101C-A, FormuMax Scientific, Sunnyvale, CA, USA) in FBS-DMEM. At 24 h after the treatment, the medium was replaced with fresh medium containing 2 µL/mL clodronate liposomes. At 24 h following the second treatment, the uterine interstitial cells were washed twice with FBS-DMEM and subsequently with HBSS and were further used for RT-PCR expression analyses to examine their properties as the niche of uterine Mø.

### 2.5. Phagocytosis Analysis of Uterine Mø with Fluorescent Beads

The phagocytic property of uterine Mø propagated by mixed culture was examined using the method of Ogawa et al. [21]. After segregation using the bacteriological Petri dish, 1 µL fluorescent yellow-green-conjugated latex beads (mean diameter, 1 µm; L4655, Sigma-Aldrich) were added to the cell suspension (2.5 × 10^5^ cells in 0.5 mL DMEM-FBS). Cells were incubated for 2 h at 37 °C with gentle shaking, washed with HBSS containing 1 mM EDTA, seeded in a 3.5 cm glass-bottom dish (3910-035, AGC Techno Glass) containing FBS-DMEM, and then incubated for approximately 2 h until almost all cells had adhered to the glass surface. After washing once with FBS-DMEM, cells were fixed with 10% formalin (Kanto Chemical, Tokyo, Japan) in PBS at 24 °C for 10 min. Subsequently, green fluorescence and phase-contrast images of the same fields were captured (IX71; Olympus, Tokyo, Japan). We regarded cells that phagocytized more than two latex beads as Mø and counted >1000 cells in each sample. The percentage of Mø in segregated cells using the bacteriological Petri dish was calculated from four independent experiments (four propagated cells from four mice uteruses). Data are represented as the mean ± SD.

### 2.6. Flow Cytometry

Flow cytometry was performed to examine the expressions of 16 Mø markers (CD11b, CD11c, CD36, CD45, CD68, CD115, CD116, CD169, CD184, CD192, CD206, arginase 1, F4/80, iNOS, Mertk, and MHC II) and 4 steroidogenic molecules (StAR, HSD3B, HSD17B1, SRD5A1) in uterine tissue-resident Mø propagated by mixed culture according to the method of Tsurutani et al. [22] with modifications. The monoclonal antibodies (Mø markers) and polyclonal antibodies (steroidogenic molecules) used in the flow cytometric analyses are listed in Appendix A, respectively. Cells at a density of ~1 × 10^6^/mL in BSA-PBS were fixed in 5% formalin for 20 min at 24°C. The cells were washed with BSA-PBS and permeabilized using 0.2% saponin (Nacalai Tesque, Kyoto, Japan) in 1 mL BSA-PBS (Sap-BSA-PBS) for 10 min at 24 °C. To avoid nonspecific Fc-gamma receptor-mediated binding of fluorochrome-conjugated antibodies, 2.5–4 × 10^5^ cells/50 µL of the cell suspensions were pretreated with 0.5 µg of anti-mouse CD16/32 antibody for 10 min at 24 °C. For the detection of Mø markers, the appropriate quantity of the antibody listed in Appendix A was added to the 50 µL cell suspension and incubated for 20 min at 24 °C. As controls, we used cell suspensions that were pretreated with the anti-mouse CD16/32 antibody and treated with the same quantity of fluorochrome-labeled isotype control antibody listed in Appendix A. For the detection of steroidogenic molecules, 0.5 µg of the primary polyclonal rabbit antibodies listed in Appendix A (anti-StAR rabbit antibody, anti-pan-HSD3B rabbit antibody, anti-HSD17B1 rabbit antibody, anti-SRD5A1 rabbit antibody) was added to the 50 µL cell suspension and incubated for 30 min at 24 °C. After washing, 1.0 µg Alexa-488-conjugated anti-rabbit or 1.0 µg PE-conjugated anti-rabbit secondary antibody (listed in Appendix A) was added to the 50 µL cell suspension and incubated for 15 min at 24 °C. As controls, we used cell suspensions that were pretreated with the anti-mouse CD16/32 antibody and treated with the same quantity of the fluorochrome-labeled secondary antibody. After washing, 20,000 cells were analyzed using a flow cytometer (CytoFLEX S, Beckman Coulter, Brea, CA, USA) for their expression characteristics. The expression of those molecules was determined from 4 independent experiments using uterine Mø derived from 8 mice.

### 2.7. Immunofluorescence Staining

The uterus was transversely cut into 2–3 mm thick sections, fixed using 10% formalin in PBS for 3–4 h on ice with gentle shaking, washed with PBS, immersed in 30% sucrose in PBS for 16 h at 4 °C, and then mounted in a compound for cryostat sections (4583, Sakura Finetechnical, Tokyo, Japan). An anti-F4/80 rat monoclonal antibody (BM8; BMA Biomedicals, Augst, Switzerland) and anti-CD206 rat monoclonal antibody (MR6F3, 17-2061-80, Thermo Fisher Scientific, Waltham, MA, USA) were used to identify tissue-resident Mø. An anti-Ki67 rabbit monoclonal antibody (SP6; NeoMarkers, Inc., Fremont, CA, USA) was used to identify proliferating cells. An anti-GJA1 (connexin 43) rabbit polyclonal antibody (C6219, Sigma-Aldrich) was used to identify gap junctions. An anti-pan-HSD3B rabbit polyclonal antibody (KO607; Trans Genic Inc., Kobe, Japan) and an anti-HSD17B1 rabbit polyclonal antibody (144-61992; RayBiotech, Peachtree Corners, GA, USA) as well as an anti-StAR rabbit polyclonal antibody (12225-1-AP; Proteintech, Rosemont, IL, USA) and anti-SRD5A1 rabbit polyclonal antibody (26001-1-AP, Proteintech) were used to identify steroidogenic cells. Double immunofluorescence staining was performed as previously described [23]. Cryostat sections (6 µm thick) were incubated with a mixture of 2 µg/mL rat anti-F4/80 and 1:100 rabbit anti-Ki67 antibody, 2 µg/mL rabbit anti-pan-HSD3B, 1:100 rabbit anti-StAR, 1:200 rabbit anti-HSD17B1, or 1:100 rabbit anti-SRD5A1 antibody as well as 1 µg/mL rat anti-CD206 and 1.5 µg/mL rabbit anti-GJA1 in BSA-PBS containing 0.02% Triton X-100 (BSA-PBS-T) for 90 min at 32 °C. After the sections were washed with PBS, they were incubated with a mixture of 5 µg/mL Alexa Fluor 488-conjugated donkey anti-rat IgG (A21208, Thermo Fisher Scientific, Waltham, MA, USA) and 5 µg/mL Alexa Fluor 594-conjugated donkey anti-rabbit IgG (711-585-152, Jackson ImmunoResearch Laboratories, West Grove, PA, USA) as well as a mixture of 5 µg/mL Alexa Fluor 488-conjugated donkey anti-rabbit IgG (A21206, Thermo Fisher) and 5 µg/mL Alexa Fluor 594-conjugated donkey anti-rat IgG (A21209, Thermo Fisher) in BSA-PBS-T for 30 min at 32 °C. To label the nucleus, some sections were stained with 4′,6-diamidino-2-phenylindole dihydrochloride (DAPI, 2 µg/mL; 342-07431, FUJIFILM Wako Pure Chemical, Tokyo, Japan), which was included in the mixture of secondary antibodies. The specificity of the staining was verified by incubation without primary antibodies. After washing and mounting with PermaFluor (TA-030-FM, Thermo Fisher), green, red, and/or blue fluorescence images of the same fields were captured under a fluorescence microscope (IX71; Olympus). 

### 2.8. RT-PCR Analyses

We examined gene expressions of steroidogenic molecules (StAR, CYP11A1, HSD3B1, HSD3B2, HSD3B6, CYP17A1, HSD17B1, HSD17B2, HSD17B3, SRD5A1, and CYP19A1); transcription factors regulating the expression of steroidogenic molecules (SF-1, LRH-1, GATA4, and DATA6); sex steroid receptors (progesterone receptor [PGR], androgen receptor [AR], estrogen receptors [ER1 and ER2]); growth factors for Mø (CSF1, CSF2, and IL34); chemokines (CCL2, CXCL9, and CXCL12); and a gap junctional molecule (GJA1/connexin 43) as well as transcription factors shaping tissue-resident Mø specificities (BACH1, BACH2, CEBPB, DTX4, ID2, ID3, LXRα, PPARγ, RUNX3, SALL1, SMAD2, SMAD3, SPIC, and PU.1) in uterine Mø and/or uterine fibroblastic cells by RT-PCR based on the following evidence: (i) the uterus is a target organ of sex steroids, and endometrial epithelial cells and stromal cells express sex steroid receptors [28,30]; (ii) sex steroids are produced de novo and/or metabolized locally in the uterus, especially in endometrial epithelial cells and stromal cells (fibroblasts) [28,31]; (iii) transcription factors such as SF-1 and GATA4 are involved in the expression of steroidogenic molecules [32,33]; (iv) cells serving as the niche of the tissue-resident Mø expressed not only growth factors for Mø but also certain chemokines [24], which are produced by endometrial stromal cells [34,35], and CXCL9 induces migration of Mø [36]; (v) endometrial stromal cells express GJA1, which is likely involved in embryonic implantation [37], and gap junctions were formed between testicular interstitial Mø and Leydig cells, which showed properties of the niche of the Mø [23]; (vi) some transcription factors characterize resident tissue/organ-specific properties in several representative tissue-resident Mø (SALL1, SMAD2, SMAD3: microglia; ID2, RUNX3: Langerhans cells; BACH2, CEBPβ, PPARγ: alveolar Mø; ID3, LXRα, SPIC: Kupffer cells; DTX4, RUNX3: intestinal Mø; and BACH1, SPIC: red pulp Mø), and PU.1 is a master regulator in Mø differentiation and development [2,3].

RT-PCR was performed as previously described [24]. Total RNA was isolated using the TRI Reagent (TR118, Molecular Research Centre, Cincinnati, OH, USA) from tissue-resident Mø and fibroblastic cells propagated by mixed culture from the uterus. Briefly, using Moroney Murine Leukemia Virus reverse transcriptase, RNase H^−^ (316-08151, Nippon Gene, Toyama, Japan), and an oligo (dT)18 primer, total RNA (1 μg) was transcribed into first-strand cDNA, and then, using the reverse-transcribed cDNA as the template, 0.5 µL of a 25 µL reaction mixture was amplified with Taq DNA polymerase (TaKaRa Ex Taq HS, RR006A; TaKaRa Bio, Otsu, Japan). The primer pairs and thermal cycling conditions used for PCR amplification are listed in Appendix A. As negative controls, the RT reaction was not performed. The PCR products were separated on 1.5% agarose gels and visualized using GelRed (41002, Biotium, Inc., Fremont, CA, USA). The gene expressions were determined from more than four independent experiments in respective tissues and cells derived from more than four mice.

## 3. Results

### 3.1. Distribution and Proliferative Property of Tissue-Resident Mø in the Uterus 

We first examined the distribution of tissue-resident Mø and their proliferative property in the uterus of four estrous cycle phases in naturally cycling mice by double immunofluorescence microscopy. F4/80-positive Mø were distributed at a high density not only in the endometrium but also in the myometrium and perimetrium of all four estrous cycle phases (Figure 1A). In contrast, Ki67-positive cells were frequently observed in the luminal epithelial cells of the proestrus, the estrus, the metestrus, and a part of the diestrus phases as well as frequently in uterine glandular cells of the proestrus phase and infrequently of the estrus phase. Ki67-positive interstitial cells appeared frequently in the endometrium of a part of the diestrus and proestrus phases but infrequently in all layers of the uterus in all four estrous cycle phases (Figure 1A). Moreover, Ki67-positive Mø occasionally appeared in all layers of the uterus, especially in the proestrus and a part of the diestrus phases (Figure 1B). Therefore, we propagated uterine-tissue-resident Mø using the mice uterus mainly from the proestrus and diestrus phases of the estrous cycle.

### 3.2. Propagation of Uterine Mø by Mixed Culture and Their Segregation 

We successfully propagated uterine-tissue-resident Mø by mixed culture with uterine fibroblastic cells from the mouse uterus using standard culture media. The propagation behaviors of uterine Mø and other uterine cells, as well as their morphologies and cell components in the mixed culture, were almost the same among mice uteruses in the four estrous cycles. Primary uterine interstitial cells, including Mø, showed high propagation, became overconfluent, and formed a multilayered structure within one week with the replacement of the culture medium every 2–3 days (Figure 2A–D). Interstitial cells showing fibroblastic morphology propagated especially rapidly in the primary culture. Overconfluent uterine cells were then sub-cultured at a dilution ratio of 1:3 until they reached overconfluence again, which usually occurred within 2 weeks (Figure 2E–H). Primary uterine cells showing epithelial morphology also engrafted on the dishes (Figure 2A,B) but did not clearly propagate in the primary culture. After the first passage, uterine Mø and other uterine cells showing fibroblastic morphology engrafted and propagated on the dishes (Figure 2E,F), whereas cells showing epithelial morphology were not clearly engrafted in the mixed culture of the sub-culture. Uterine Mø propagated along with the propagation of uterine fibroblastic cells in the sub-culture of the first passage. Mø were morphologically identified as small flat cells with a few cytoplasmic protrusions as well as small round or fusiform cells in the mixed culture (Figure 2).

The interstitial cells, composed mainly of uterine Mø and fibroblastic cells, could be sub-cultured for more than three passages, whereas their propagation in the mixed culture became slow with repeating sub-cultures: in the sub-culture of the third passage, uterine Mø propagated far slower along with the slow propagation of fibroblastic cells. Therefore, the overconfluent cells of the first and second passage were used for subsequent analyses. Overconfluent primary cells were frozen at a dilution ratio of 1:3. Frozen cells were treated under the same cultivation conditions and found to propagate somewhat slowly compared to the unfrozen cells. 

As Mø can exclusively adhere to a bacteriological Petri dish without protein coating, we segregated uterine Mø from the other interstitial cells in the mixed culture using their different properties of adhering to the dish. When overconfluent uterine cells of 1–2 passages were seeded on the bacteriological Petri dish, small round/fusiform cells with a few cytoplasmic protrusions, identified as Mø, adhered to the dish surface after several hours of seeding, with evidence of cell aggregates that consisted of fibroblastic cells as a major component and Mø as a minor component, and floated in the media. These cell aggregates were easily eliminated by washing with the conditioned media (Figure 3A,B). We generally collected ~1.5 × 10^6^ adherent cells per overconfluent cell collected from a 10 cm tissue culture dish. The percentage of Mø in the separated cells that exclusively adhered to the Petri dishes was precisely determined by calculating cells phagocytizing fluorescent beads with a mean diameter of 1 µm. Almost all cells that adhered to the dish phagocytized the fluorescent beads during incubation for 2 h, and most cells phagocytized numerous beads in their cytoplasm, indicating their high phagocytic property (Figure 3C,D). We counted the bead-positive and bead-negative cells to estimate the percentage of Mø among the segregated cells. We defined cells phagocytizing more than two beads as bead-positive cells and counted >1000 cells in each sample. Bead-positive cells were 99.5 ± 0.3% of the entire population (Figure 3E), and thus almost all segregated cells comprised Mø. This indicates that the Mø separation by the adhesion property on the bacteriological Petri dish is a reliable method to obtain highly purified uterine Mø from other uterine stromal cells propagated in mixed culture. 

### 3.3. Expression Profiles of Mø Markers and Those of Transcription Factors Shaping Mø in Propagated Uterine Mø

We examined the expressions of twelve Mø markers in propagated uterine Mø by flow cytometry. Expression patterns of the markers were the same between uterine Mø derived from the uteruses of the proliferative and secretory phases (corresponding to the proestrus and diestrus phases of the estrous cycles, respectively). Histograms of the marker expression distributions showed a single fraction in propagated uterine Mø, except for those of CD36 and arginase 1. Uterine Mø revealed high expression of CD11b, CD45, CD68, CD169, CD206, and Mertk; clear expression of CD11c and F4/80; and no expression of iNOS and MHC II (Figure 4A). Moreover, a high and low expression fraction were noted in CD36 and arginase 1. Next, we examined the mRNA expression of transcription factors characterizing resident tissue/organ-specific properties in several representative tissue-resident Mø and a lineage-determining transcription factor of Mø, PU.1 in propagated uterine Mø. We found that the uterine Mø expressed *Bach1*, *Dtx4*, *Id2*, *Id3*, *Lxra*, *Pparg*, *Runx3*, *Smad2*, *Smad3*, and *Pu.1*, whereas *Runx3* and *Smad3* expressions were low as assessed by RT-PCR (32 cycles; Figure 4B).

### 3.4. Properties of Uterine Fibroblastic Cells as the Niche of Residence in Uterine Mø 

We presumed that the propagated fibroblastic cells serve as the niche for shaping uterine Mø, and thus we segregated the fibroblastic cells from uterine Mø in the mixed culture and examined the gene expressions of molecules serving for Mø propagation and attraction essential for the niche. After the treatment with clodronate-encapsulating liposomes, almost all small round/fusiform cells indicating Mø disappeared in the mixed culture, as assessed under microscopy (Figure 5A). The mixed-cultured fibroblastic cells treated with the liposomes did not express *Pu.1* (a lineage-determining transcription factor of Mø).

They expressed *Csf1*, *Csf2*, and *Il34*, all of which induce the proliferation of tissue-resident Mø, while *Csf2* and *Il34* expression was very faint as assessed by RT-PCR (32 cycles; Figure 5B). Next, we examined the gene expression of chemokines in propagated fibroblastic cells and found that they expressed *Ccl2* and *Cxcl12* (Figure 5B). Our findings likely indicate that uterine fibroblastic cells in mixed culture serve as the niche of uterine Mø due to the expression of those Mø growth factors and these chemokines. Furthermore, we examined the protein expressions of the corresponding Mø growth factor and chemokine receptors in propagated uterine Mø by flow cytometry and found clear expressions of CD115 (CSF1R) and CD116 (CSF2RA), as well as CD184 (CXCR4) and CD192 (CCR2), in the Mø (Figure 5C). 

We presumed that the formation of gap junctions connecting tissue-resident Mø is an essential property in the niche cells, and thus we examined the gene expression of GJA1 in the uterine fibroblastic cells and Mø propagated by mixed culture and found that both cells clearly express *Gja1*. Moreover, GJA1 immunoreactivity was localized between uterine Mø and interstitial cells other than Mø in the endometrium (Figure 5D). 

### 3.5. Expression of Sex-Steroid-Related Molecules in Propagated Uterine Mø and Fibroblastic Cells 

We hypothesized that tissue-resident Mø colonizing steroidogenic organs possess steroidogenic properties when they colonize among steroidogenic cells that serve as the Mø niche. Therefore, we examined the expression of molecules essential for synthesizing sex steroids listed in Figure 6A by RT-PCR. We noted *Star*, *Cyp11a1*, *Hsd3b1*, *Hsd3b6*, *Hsd17b1, Hsd17b2*, *Srd5a1*, and *Cyp19a1* expressions in uterine fibroblastic cells propagated in the mixed culture, while *Cyp11a1*, *Hsd3b1*, *Hsd3b6*, *Hsd17b1*, *Hsd17b2*, and *Cyp19a1* expressions were weak/faint as assessed by RT-PCR (32 cycles; Figure 6B). Moreover, *Star*, *Hsd3b6*, *Hsd17b1*, and *Srd5a1* were expressed in propagated uterine Mø while *Hsd3b6* and *Hsd17b1* expression were faint/weak (Figure 6B). Next, we examined the gene expression of transcription factors involved in the expression of steroidogenic molecules and receptors for sex steroids: propagated fibroblastic cells clearly expressed *Gata4* and *Gata6* and faintly expressed *Lrh-1*; propagated Mø faintly/weakly expressed *Gata4* and *Gata6;* and uterine fibroblastic cells and Mø expressed *Pgr*, *Ar*, and *Er1* (Figure 6B). Moreover, we examined the protein expression of StAR, HSD3B, HSD17B1, and SRD5A1 in propagated uterine Mø by flow cytometry. We used a pan-HSD3B antibody because we could not find an HSD3B6 antibody applicable to flow cytometry. These steroidogenic molecules were expressed in the Mø (Figure 6C). Our findings suggest that uterine Mø likely possess an ability to receive signals of sex steroids and an ability to metabolize sex steroids. 

### 3.6. Localization of StAR, HSD3B, HSD17B1, and SRD5A1 in Tissue-Resident Mø of the Uterus

We found StAR, HSD3B, HSD17B1, and SRD5A1 expression in uterine Mø propagated by mixed culture with uterine fibroblastic cells, and thus we immunohistochemically examined the localization of these molecules in the tissue-resident Mø of the uterus. We found that StAR immunoreactivity was clearly localized in F4/80-positive uterine Mø located in the endometrium, myometrium, and perimetrium (Figure 7A). StAR was also expressed in epithelial cells lining the lumen and uterine glands, as well as stromal cells in the endometrium. HSD3B and HSD17B1 immunoreactivity was clearly localized in F4/80-positive uterine Mø located similarly in all three layers of the uterus and endometrial stromal cells (Figure 7B and Figure 8A). Epithelial cells of the lumen and uterine glands as well as smooth muscle cells in the myometrium expressed HSD17B1. Moreover, SRD5A1 immunoreactivity, which appeared in the nucleus and cytoplasm, was also clearly localized in F4/80-positive uterine Mø located in all three layers of the uterus, as well as luminal epithelia, uterine glands, endometrial stromal cells, and myometrial smooth muscle cells (Figure 8B). These findings are likely the first evidence of StAR, HSD3B, HSD17B1, and SRD5A1 expression in uterine-tissue-resident Mø, to the best of our knowledge.

## 4. Discussion

### 4.1. Propagation of Uterine Mø in Mixed Culture

To the best of our knowledge, the colonization of tissue-resident Mø derived from fetal Mø, possessing a self-renewal property, in the adult uterus is not clear. Populations of leukocytes including Mø in the endometrium fluctuate in humans during the naturally cycling menstrual cycle phases [4,5] and in rodents during the estrus cycle phases [7,8]. Uterine Mø are regularly replenished by BMMs at steady state in adults [4,9], and thus uterine-tissue-resident Mø in adults are assumed to be derived totally from BMMs without studies to track and trace Mø during development and adulthood in the uterus using a fate-mapping technique such as those used in other tissue-resident Mø [38,39]. However, Mø propagate locally in the uterus [40], and their proliferation depends on CSF1 concentrations in the uterus [25]. In the present study, we immunohistochemically demonstrated that uterine-tissue-resident Mø located in the endometrium, myometrium, and perimetrium were partially Ki67-positive, suggesting cell proliferation in situ. This finding is supported by previous studies. Thus, Mø colonizing in the uterus possess an ability to propagate locally at steady state. 

Recently, we developed a simple propagation method of tissue-resident Mø applicable for Mø colonizing in the lung [22], liver [24], and testis [23] as well as the spleen and brain of adult rodents [21]. By using this method with modifications, we successfully propagated uterine-tissue-resident Mø by mixed culture with uterine interstitial cells showing fibroblastic morphology in the standard culture medium without additional supplements such as CSF1. We usually collected ~4.5 × 10^6^ uterine Mø from 2/3 tissues of the whole uterus of the adult mouse from which three 10 cm tissue culture dishes with overconfluent uterine cells were obtained in the cultivation for less than 3 weeks when primary uterine cells in the dish were sub-cultured once at a dilution ratio of 1:3. The propagated uterine Mø demonstrated a high phagocytic property, which is identical to the nature of uterine Mø because they show a high clearance property of apoptotic cells and tissue debris [9,10]. High expressions of iNOS, MHC II, and CD86 are markers of the M1 Mø phenotype, and high expressions of arginase 1, CD163, and CD206 are markers of the M2 Mø phenotype [41,42]. Previous ex vivo studies by flow cytometry using Mø/monocyte marker molecules showed that two subpopulations are present in the following CD11b-positive/F4/80-positive Mø of the mouse uterus: MHC-II-positive and MHC-II-negative/weakly positive Mø [8,25]. In contrast, studies showed that normal human endometrial Mø have properties of the alternatively activated M2 phenotype irrespective of menstrual cycles [5,12]: the major Mø population express CD11b, CD11c, CD68, CD86, CD206, and IL10 [43]; a majority of cycling endometrial Mø are CD163- and CD206-positive [44]; and the Mø are CD163-positive and MHC-II-negative/weakly positive [45]. In the present study, we showed that propagated uterine Mø have the same characteristic expression properties of Mø/monocyte marker proteins among the estrous cycle phases. Histograms displaying expression of these marker molecules consisted of a single fraction except those of CD36 and arginase 1 consisting of a low and high expression fraction: high expression of CD11b, CD36, CD45, CD68, CD169, CD184, CD206, arginase 1, and Mertk; clear expression of CD11c, CD116, CD184, CD192, and F4/80; and negative expression of iNOS and MHC II. The expression pattern of these molecules indicates that propagated uterine Mø are a typical M2 phenotype and consist of a fairly homogeneous population. In the mixed culture, uterine Mø were propagated along with the propagation of uterine fibroblastic cells in the standard medium without regular fluctuation in concentrations of sex steroids unlike the uterus in vivo under the natural menstrual/estrus cycle. Thus, propagated uterine Mø likely consisted of a homogeneous population with an M2 phenotype. This feature may be beneficial for future studies to examine uterine Mø in terms of physiology and pathology using the propagated uterine Mø because uterine Mø are likely implicated in the repair of endometrium during the natural menstrual cycle [9,10] as well as the remodeling of decidual tissues and the maintenance of immune tolerance in normal pregnancy [5,11], all of which are roles assigned to the M2 phenotype.

According to the niche of Mø residence concept, the niche shapes tissue-resident Mø to possess tissue/organ-specific properties by providing nourishment, signal molecules, and cytokines to the Mø, as well as providing CSF1, CSF2, and/or IL34 to the Mø to perform/maintain self-renewal [2,3]. Uterine Mø propagated in the mixed culture along with the propagation of uterine interstitial cells showing fibroblastic morphology. This phenomenon is likely a sign that propagation of the niche cells increases the capacity to promote Mø propagation in culture. We observed that the fibroblastic cells clearly and faintly expressed *Csf1* and *Csf2* and *Il34*, respectively, as well as clearly expressed *Ccl2* and *Cxcl12*. Propagated uterine Mø expressed corresponding cytokine receptors (CD115 [CSF1R] against CSF1 and IL34; CD116 [CSF2RA] against CSF2) and chemokine receptors (CD184 [CXCR4] against CXCL12; CD192 [CCR2] against CCL2). Therefore, the CSF1/CSF1R interaction likely induced the propagation of uterine Mø in the mixed culture, as shown in a previous report [46]. Moreover, propagated uterine fibroblastic cells and Mø clearly expressed *Gja1*, and GJA1 immunoreactivity was localized between Mø and interstitial cells in the endometrium, suggesting gap junction formation between endometrial Mø and stromal cells. Endometrial stromal cells produce CSF1 [27], CCL2 [34], and CXCL12 [35], as well as express GJA1 [37]. Propagated fibroblastic cells possess an identical property of endometrial stromal cells (fibroblasts) in terms of these molecular expressions, and thus endometrial stromal cells are a suitable candidate for the niche of uterine-tissue-resident Mø colonizing in the endometrium. Signals by CSF1 and CSF2 in Mø induce the polarization of Mø toward the M2 and M1 phenotype, respectively [47]. Therefore, the high expression of *Csf1* in the fibroblastic cells (endometrial stromal cells) is possibly implicated in the maintenance of the M2 polarization property in the propagated uterine Mø and in the endometrial Mø. Moreover, chemokine receptor signals in leukocytes induce cell–cell adhesion via integrin activation [48,49]. Therefore, CCL2-CCR2 and CXCL12-CCR4 interaction are possibly involved in cell–cell adhesion between endometrial stromal cells and Mø, which are likely functionally coupled via the formation of gap junctions between the two cells. Further studies are required to elucidate these speculations. 

### 4.2. Metabolic Properties of Sex Steroids in Uterine Mø

The uterus is a target organ for sex steroids from the ovary and adrenal gland [30,50]. PGR, AR, and ER1 are expressed in endometrial epithelial and stromal cells [51,52], whereas PGR and ER expression in endometrial Mø are controversial: one study showed PGR and ER expressions in human endometrial Mø [53], but another study showed their negative expressions in human endometrial leukocytes, including Mø [54]. By contrast, accumulated evidence has shown that sex steroids are locally produced/metabolized in the uterus: genes and/or proteins of steroidogenic molecules such as StAR, CYP11A1, HSD3B1, HSD3B2, CYP17A1, HSD17B1, HSD17B2, CYP19A1, and SRD5A1 are expressed in endometrial stromal cells and/or epithelial cells [28,31]. Recently, we found that testicular interstitial Mø propagated by mixed culture with fibroblastic cells showing properties of Leydig cells in culture produce progesterone de novo, whereas propagated liver, lung, and spleen Mø did not express steroidogenic molecules [23]. Based on the evidence, we hypothesized that tissue-resident Mø colonizing around sex-steroid-producing cells such as endometrial stromal cells have the ability to produce sex steroids de novo and/or convert/metabolize sex steroids. We found that uterine fibroblastic cells propagated in the mixed culture expressed *Star*, *Cyp11a1*, *Hsd3b1*, *Hsd3b6*, *Hsd17b1*, *Hsd17b2*, *Srd5a1*, and *Cyp19a1* and that uterine Mø propagated in the mixed culture expressed *Star*, *Hsd3b6*, *Hsd17b1*, and *Srd5a1*. Both propagated cells expressed sex steroid receptors, *Pgr*, *Ar*, and *Er1*. They also expressed *Gata4* and *Gata6* and *Lrh-1* additionally in fibroblastic cells, all of which are transcription factors upregulating expression of steroidogenic molecules [32,33]. Moreover, StAR, HSD3B, HSD17B1, and SRD5A1 were expressed in the propagated uterine Mø and immunohistochemically localized in tissue-resident Mø located in the endometrium, myometrium, and perimetrium. Therefore, (i) propagated fibroblastic cells possess steroidogenic properties and are likely identical to endometrial stromal cells (fibroblasts), which likely indicates that fibroblastic cells in the mixed culture with uterine Mø maintain an essential property of endometrial stromal cells; (ii) propagated uterine Mø possess sex steroid metabolic properties but not sex-steroid-producing properties de novo due to the non-expression of CYP11A1, which is an essential enzyme in the initial step of steroidogenesis to convert cholesterol to pregnenolone [55]; and (iii) not only progesterone and estrogen but also androgen likely target uterine Mø in addition to endometrial stromal cells. Our findings and evidence from previous studies are collectively summarized in Figure 9. 

Propagated uterine Mø expressed CD36, a scavenger receptor with affinity to oxidized low-density lipoproteins [56], as well as *Lxra* and *Pparg*, transcription factors regulating lipid metabolism [57,58]. StAR is a cholesterol transport protein regulating steroidogenesis in the rate-limiting initial step [59], and thus uterine Mø may be deeply implicated in cholesterol metabolism. Moreover, *Gja1* was expressed in propagated uterine Mø and fibroblastic cells. GJA1 immunoreactivity was localized between Mø and other interstitial cells in the endometrium, suggesting gap junction formation between endometrial Mø and stromal cells. Thus, these two cells may be functionally coupled with each other via gap junction channels permeable to small molecules including metabolites and second messengers, such as cAMP [60], and thus may concertedly perform functions to establish an endometrial environment suitable for pregnancy. Further studies are required to elucidate these speculations.

SRD5A1 immunoreactivity appeared in the nucleus and cytoplasm while StAR, HSD3B, and HSD17B1 immunoreactivity appeared in the cytoplasm. The nuclear and cytoplasmic expression of SRD5A1 was shown in a previous study [61] and thus the cellular localization of SRD5A1 was different from that of the other steroidogenic molecules. 

Dihydrotestosterone (DHT) is the most potent endogenous androgen that is converted from testosterone by SRD5A1 [62]. Androgen performs intracrine action in preparation of the uterine tissue environment to support implantation and establishment of pregnancy, as well as in the etiology of disorders such as endometriosis [28,30,31]. We found SRD5A1 expression in propagated uterine Mø and SRD5A1 immunoreactivity in the tissue-resident Mø located in the endometrium, myometrium, and perimetrium. Thus, uterine Mø may modulate androgen intracrine action by converting testosterone to more potent DHT. Further studies are required in this regard. 

## 5. Conclusions

We successfully propagated tissue-resident Mø by mixed culture from mouse uterine cells in standard culture medium. The uterine Mø propagated along with the propagation of fibroblastic cells, showing properties of endometrial stromal cells such as the expression of steroidogenic molecules and sex steroid receptors. The fibroblastic cells also exhibited properties of the niche of Mø, such as the expression of CSF1 to promote proliferation of Mø and certain chemokines to induce cell–cell adhesion via integrin activation. Propagated uterine Mø were CD206- and arginase-1-positive; iNOS- and MHC-II-negative, indicating the polarization of the M2 phenotype; and highly phagocytic. These properties are identical to those of endometrial Mø. Thus, the niche of uterine Mø can be successfully reproduced in vitro, and this is the first study to establish the propagation method of uterine Mø in culture. We newly identified that (i) uterine Mø expressed steroidogenic molecules, including SRD5A1 converting testosterone to DHT, and (ii) uterine Mø expressed *Gja1*, and GJA1 was localized between endometrial uterine Mø and other interstitial cells, suggesting gap junction formation between the Mø and endometrial stromal cells. As androgen performs intracrine action in preparation of the uterine tissue environment to support implantation and establishment of pregnancy, as well as in the etiology of disorders such as endometriosis, uterine Mø may be a new tool leading to further studies on immune–endocrine interactions in the uterus related to fertility and infertility.

## Figures and Tables

**Figure 1 biomedicines-11-00985-f001:**
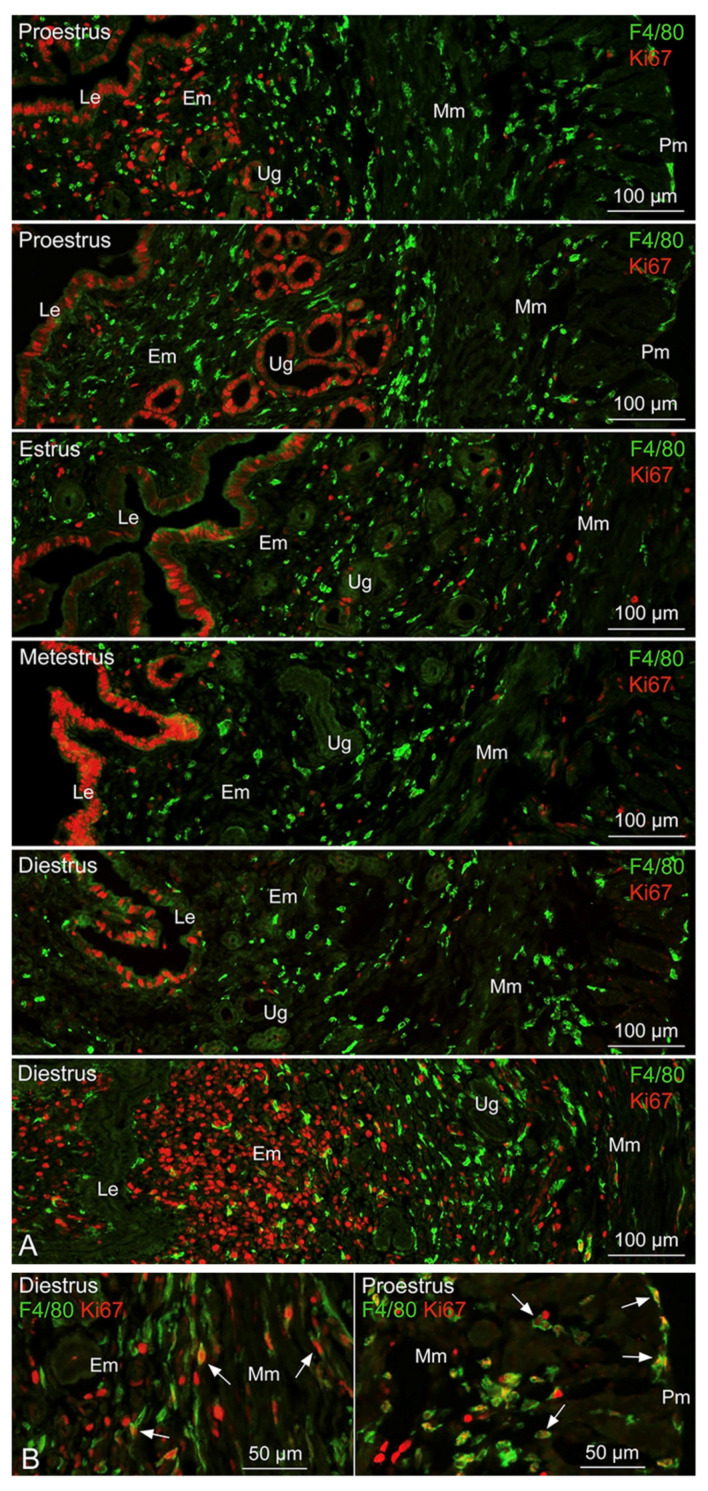
Double immunofluorescence micrographs showing the localization of F4/80-positive Mø (green) and Ki67-positive proliferating cells (red) in the uterus in the four estrus cycle phases (proestrus, estrus, metestrus, and diestrus) of naturally cycling mice. Green fluorescence images were merged with red fluorescence images in the same field. (**A**) F4/80-positive Mø are distributed at a high density in the endometrium (Em) and myometrium (Mm) as well as perimetrium (Pm) of all four estrous cycle phases. (**B**) F4/80-positive Mø are occasionally Ki67-positive (arrows) in all layers of the uterus of proestrus and diestrus phases. Le, luminal epithelium; Ug, uterine gland.

**Figure 2 biomedicines-11-00985-f002:**
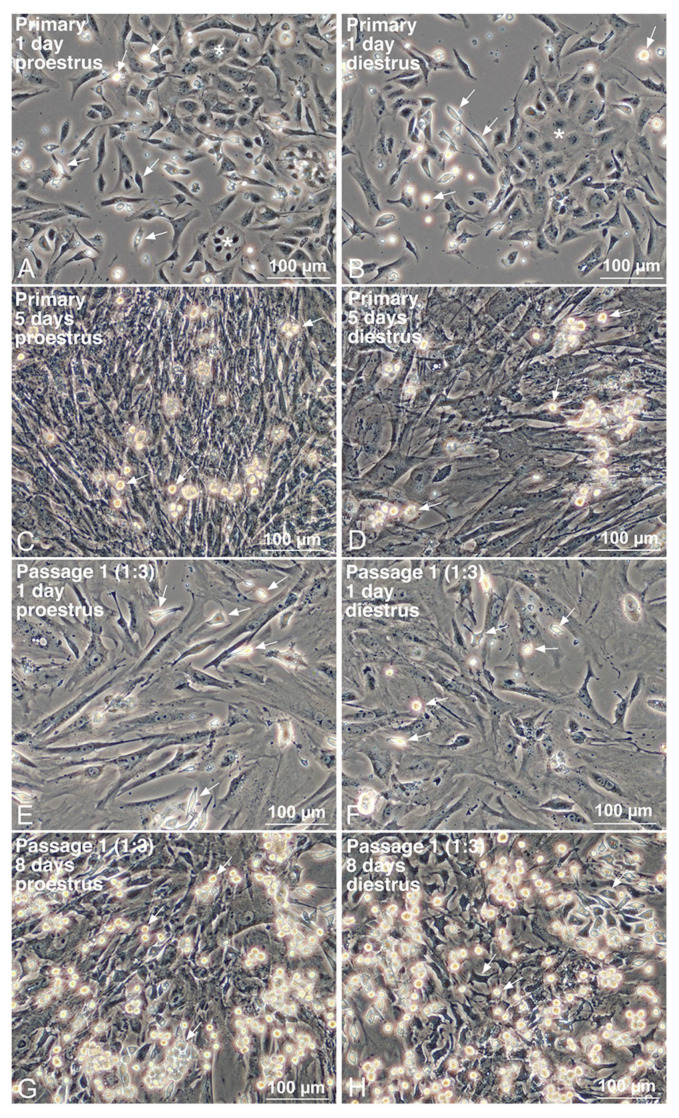
Uterine-tissue-resident Mø propagation along with the propagation of uterine fibroblastic cells in the mixed culture. Cells from the mouse uterus of the proestrus (**A**,**C**,**E**,**G**) and diestrus phases (**B**,**D**,**F**,**H**) were mixed primary-cultured and sub-cultured for the indicated days. Typical phase-contrast micrographs indicating the propagation of uterine-tissue-resident Mø in the primary culture (**A**–**D**) and sub-culture (**E**–**H**) by mixed culturing with uterine cells in tissue culture dishes. Arrows, Mø; *, cobblestone-like structure (epithelial cells).

**Figure 3 biomedicines-11-00985-f003:**
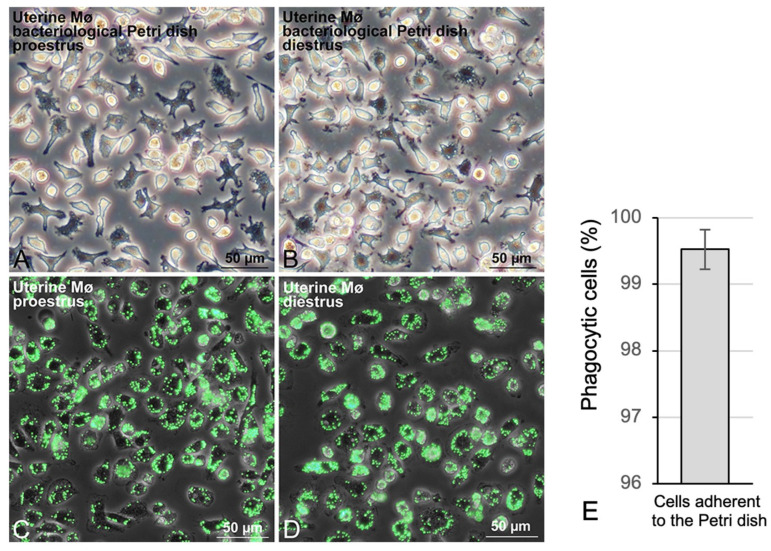
Uterine Mø segregated from uterine fibroblastic cells in mixed culture by using their differing property of adhering to bacteriological Petri dishes. (**A**,**B**) Typical phase-contrast images showing tissue-resident Mø derived from the mouse uterus in the proestrus (**A**) and diestrus phase (**B**) of the estrus cycle corresponding to the proliferating and secretory phase of the uterus, respectively, in bacteriological Petri dishes. Nonadherent cells and cell aggregates were removed by washing with conditioned media. Mø selectively adhered to the surface of the Petri dish. (**C**,**D**) Typical florescence images merged with phase-contrast images showing phagocytic properties assessed by incubation with fluorescent beads in propagated tissue-resident Mø derived from the mouse uterus in the proestrus (**C**) and diestrus phase (**D**) and segregated by adhesion to bacteriological Petri dishes. Most cells contain numerous beads in their cytoplasm. (**E**) A bar graph showing the percentage of uterine Mø phagocytizing beads in cells adherent to the Petri dishes that is presented as mean ± SD. More than 1000 cells per sample were counted, and the percentage of cells phagocytizing beads was determined from 4 independent experiments in 4 mice.

**Figure 4 biomedicines-11-00985-f004:**
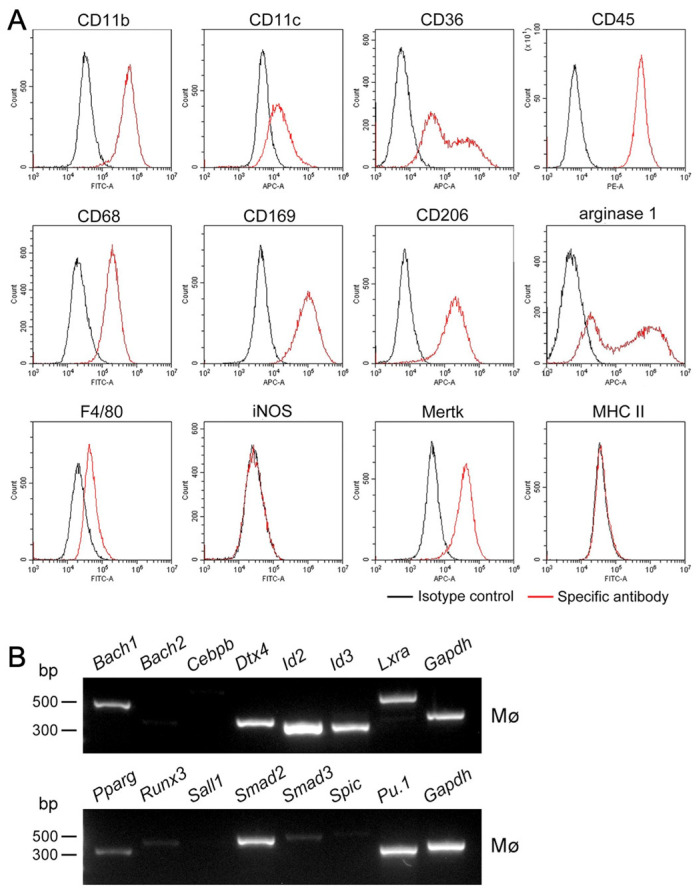
Expression profiles of Mø markers and transcription factors in propagated uterine Mø. **(A**) Typical histograms showing protein expression of Mø markers examined by flow cytometry. Note that uterine Mø are CD206- and arginase-1-positive as well as iNOS- and MHC-II-negative, indicating the polarization of the M2 phenotype. Red histogram, specific antibody; black histogram, isotype control. (**B**) Representative mRNA expression profiles of transcription factors shaping resident tissue/organ-specific identities of the representative tissue-resident Mø examined by RT-PCR.

**Figure 5 biomedicines-11-00985-f005:**
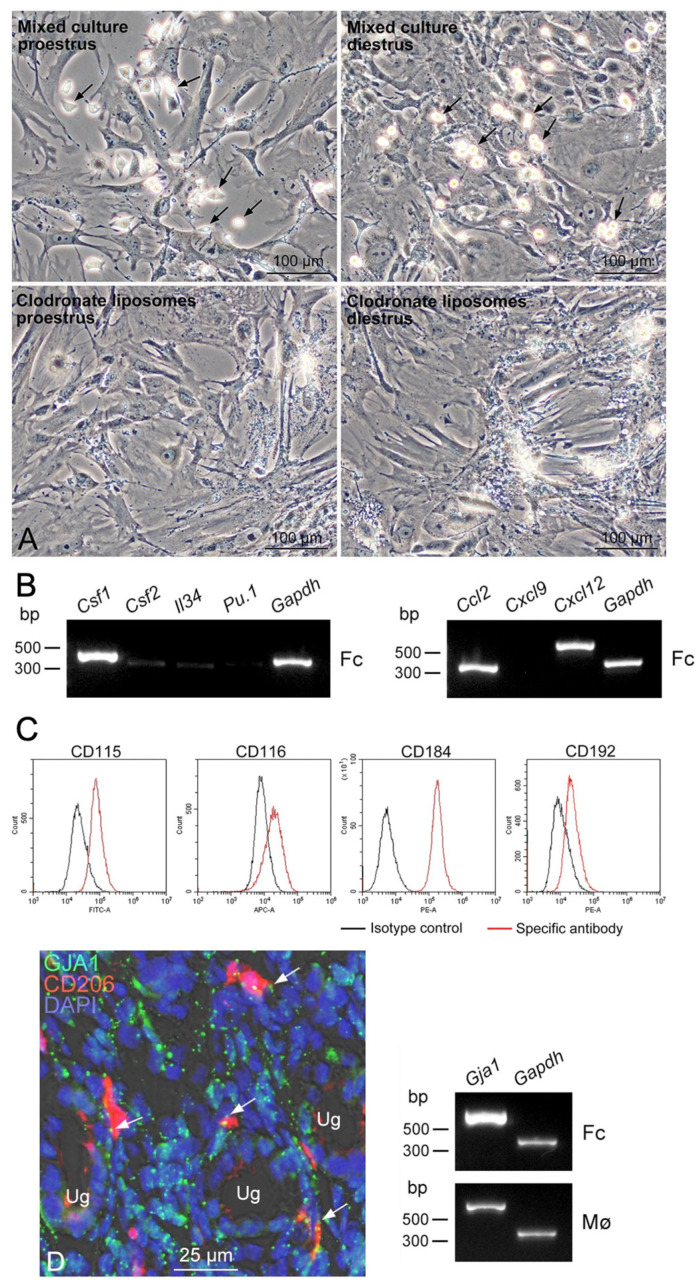
Mø niche properties in uterine fibroblastic cells propagated by mixed culture. Uterine fibroblastic cells were propagated by mixed culture with uterine Mø, treated with clodronate-encapsulating liposomes to deplete uterine Mø, and used as samples to examine the expression of molecules related to the Mø niche by RT-PCR. (**A**) Typical phase-contrast images of uterine fibroblastic cells propagated in the mixed culture from the uterus of the indicated estrus phases and treated with (lower panels) or without the clodronate liposomes (upper panels). Arrows, Mø. (**B**) Representative mRNA expression profiles of molecules related to the Mø niche. mRNA expression in fibroblastic cells (Fc) treated with the liposomes was analyzed using RT-PCR. (**C**) Representative protein expression of CD115, CD116, CD184, and CD192 determined by flow cytometry in propagated uterine Mø. Red-line histogram, specific antibody; black-line histogram, isotype control. (**D**) Representative mRNA expression of GJA1 in propagated fibroblastic cells (Fc) and Mø by RT-PCR, and immunofluorescence image merged with phase-contrast image showing gap junction localization between uterine Mø and other interstitial cells in the endometrium. Punctate GJA1 immunoreactivity (green) is localized in CD206-positive (red) Mø and adjacent interstitial cells (arrows). Ug, uterine gland.

**Figure 6 biomedicines-11-00985-f006:**
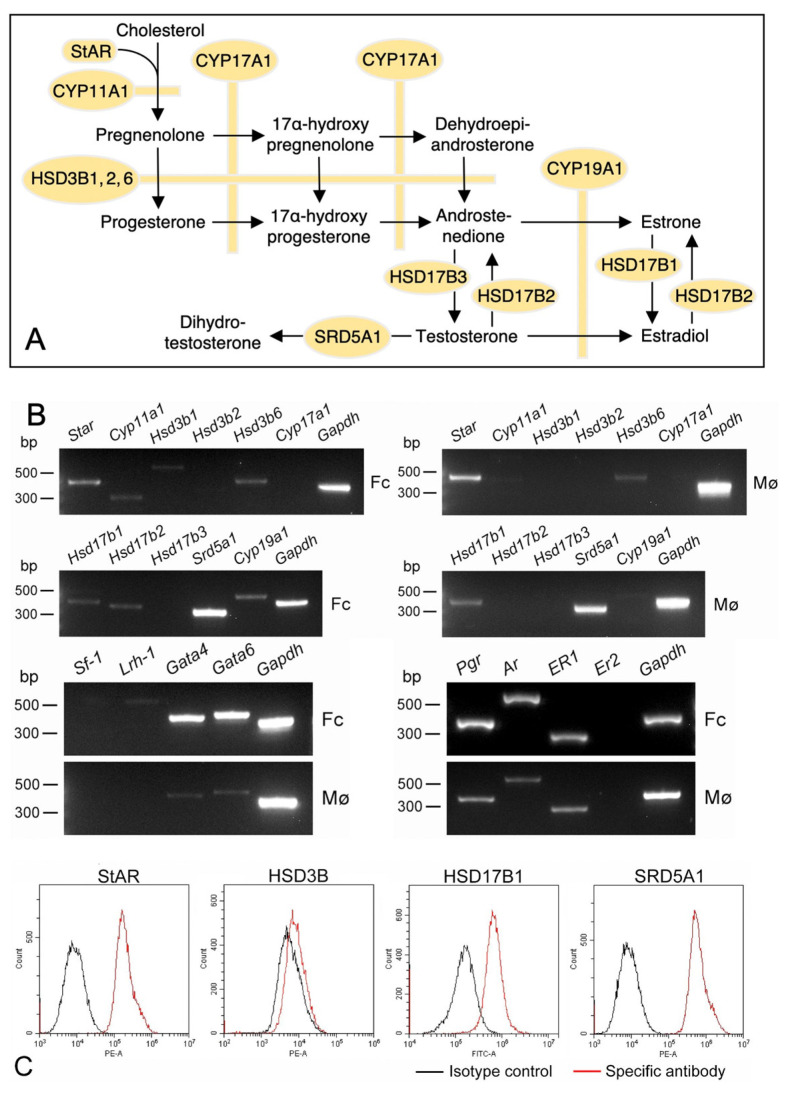
Expression properties of sex-steroid-related molecules in uterine fibroblastic cells and Mø. (**A**) Schematic drawing showing pathways and molecules involved in sex steroid synthesis from cholesterol. (**B**) Representative mRNA expressions of steroidogenic molecules, transcription factors upregulating steroidogenic molecules, and sex steroid receptors in propagated uterine fibroblastic cells (Fc) and Mø. (**C**) Representative histograms displaying protein expression of StAR, HSD3B, HSD17B1, and SRD5A1 determined by flow cytometry in propagated uterine Mø. Red-line histogram, specific antibody; black-line histogram, isotype control.

**Figure 7 biomedicines-11-00985-f007:**
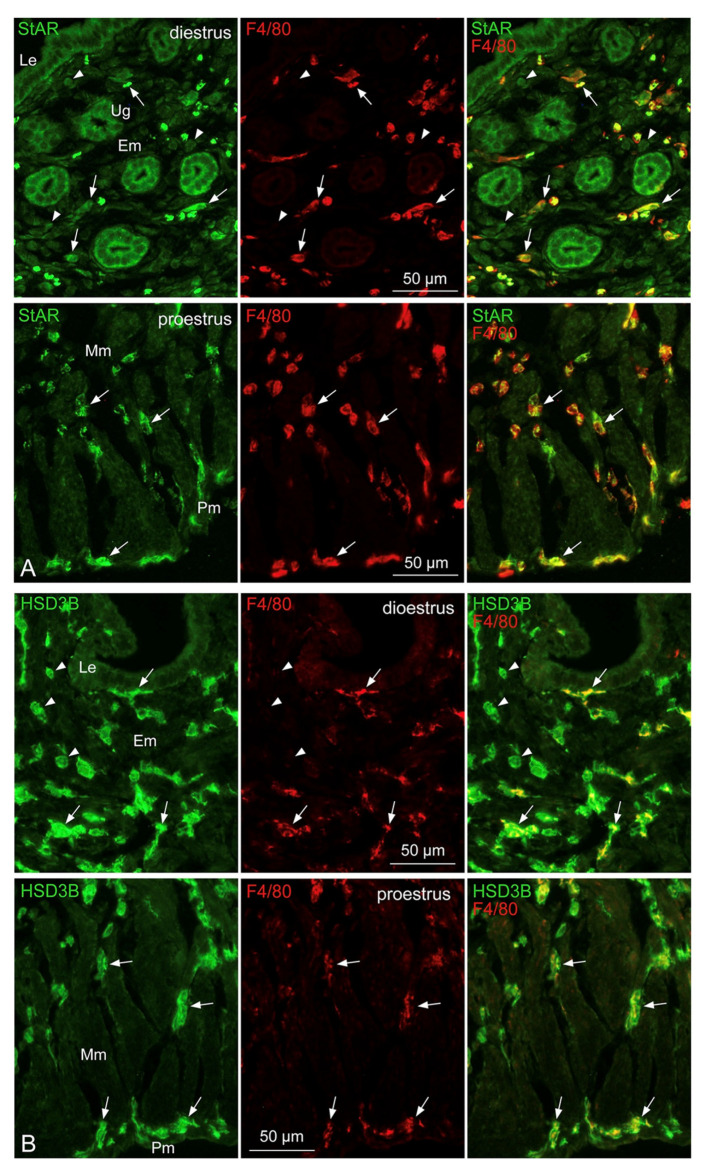
Immunohistochemical localization of StAR and HSD3B in uterine Mø. Representative double immunofluorescence micrographs showing StAR- (green) and F4/80-positive Mø (red) as well as HSD3B- (green) and F4/80-positive Mø (red) in the uterus of proestrus and diestrus phases in naturally cycling mice. Green fluorescence images were merged with red fluorescence images in the same field (right panel). (**A**,**B**) StAR (A) and HSD3B (B) are clearly expressed in Mø (arrows) located in the endometrium (Em), myometrium (Mm), and perimetrium (Pm) as well as endometrial stromal cells (arrowheads). StAR is also clearly expressed in luminal epithelial cells (Le) and uterine glandular cells (Ug).

**Figure 8 biomedicines-11-00985-f008:**
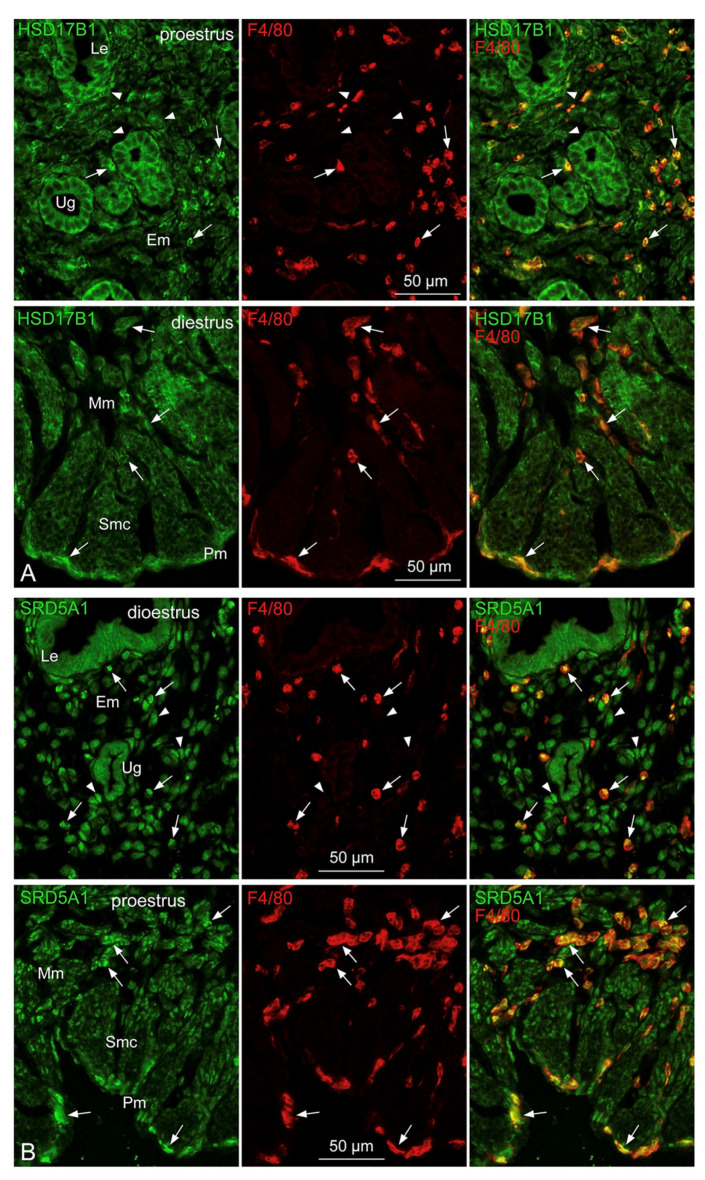
Immunohistochemical localization of HSD17B1 and SRD5A1 in uterine Mø. A: Representative double immunofluorescence micrographs showing HSD17B1- (green) and F4/80-positive Mø (red) as well as SDR5A1- (green) and F4/80-positive Mø (red) in the uterus of proestrus and diestrus phases in naturally cycling mice. Green fluorescence images were merged with red fluorescence images in the same field (right panel). (**A**,**B**) HSD17B1 and SDR5A1 immunoreactivity appears in Mø (arrows) located in the endometrium (Em), myometrium (Mm), and perimetrium (Pm) as well as luminal epithelial cells (Le) and uterine glandular cells (Ug), endometrial stromal cells (arrowheads), and smooth muscle cells (Smc), while SDR5A1 immunoreactivity is localized in the cytoplasm and also the nucleus.

**Figure 9 biomedicines-11-00985-f009:**
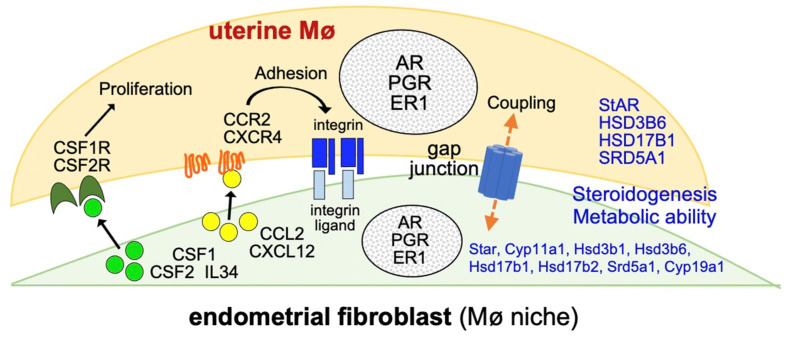
A schematic drawing deduced from the present findings showing steroidogenic properties in tissue-resident Mø that colonize among endometrial stromal cells serving as the niche of Mø residence.

## Data Availability

The data presented in this study are available upon request from the corresponding author.

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
