# Peer review of "Mixed-Culture Propagation of Uterine-Tissue-Resident Macrophages and Their Expression Properties of Steroidogenic Molecules"

_biomedicines, 2023, doi:10.3390/biomedicines11030985_

Round 1
Reviewer 1 Report
Comments to the Authors
This is an article on uterine tissue-resident macro-phages culture and proliferation and their steroidogenic characteristics. While the manuscript has been carefully written, I would like to suggest some modifications to improve its quality:
- Some sentences should be added to the introduction section about the interaction between endometrium stromal cells and M2(Mø) in endometriosis (line 49). These articles can be helpful Possible involvement of signal transducer and activator of transcription-3 in cell–cell interactions of peritoneal macrophages and endometrial stromal cells in human endometriosis - ScienceDirect and Macrophages Are Alternatively Activated in Patients with Endometriosis and Required for Growth and Vascularization of Lesions in a Mouse Model of Disease - ScienceDirect
- There are many repetitions in the whole text for example” We recently demonstrated that testicular interstitial Mø”. The text should be rechecked, and repetitions should be omitted.
- In the result section, the citation of other studies should be removed. If needed discuss them in the discussion section. For example, in line, 373.
- In figure 3 similar magnification should be used.
- In flow cytometry figure " Figure 4 " the percentage of different positive cells for each antibody should be mentioned.
- In the discussion section line, 538 references should be cited.
Author Response
Responses to the comments of the Reviewers
We have revised the manuscript to address the comments of Reviewer 1 as outlined below. We have also highlighted the changes within the document in blue.
Reviewer 1
Comment 1: Some sentences should be added to the introduction section about the interaction between endometrium stromal cells and M2(Mø) in endometriosis (line 49). These articles can be helpful: Possible involvement of signal transducer and activator of transcription-3 in cell-cell interactions of peritoneal macrophages and endometrial stromal cells in human endometriosis - ScienceDirect and Macrophages Are Alternatively Activated in Patients with Endometriosis and Required for Growth and Vascularization of Lesions in a Mouse Model of Disease - ScienceDirect
Response: Thank you for the reviewer’s comment. According to the comments we added some sentences and cited the two papers suggested by the reviewer. This is now in lines 49-52 on page 2.
Comment 2: There are many repetitions in the whole text for example” We recently demonstrated that testicular interstitial Mø”. The text should be rechecked, and repetitions should be omitted.
Response: Thank you for the comment. We checked repetitions in the original manuscript and removed/rewrote sentences/phrases of repetitions in the revised manuscript as much as possible.
Comment 3: In the result section, the citation of other studies should be removed. If needed discuss them in the discussion section. For example, in line, 373.
Response: Thank you for the comment. We removed the sentences with references in the Result section and moved many of them to the Materials/Methods and Discussion section: these are now in lines 244-258 on pages 5 and 6 and in lines 613-616 on page 18.
Comment 4: In figure 3 similar magnification should be used.
Response: Thank you for the comment. We replaced the upper panels of the micrographs (Fig. 3A, B) in the revised manuscript.
Comment 5: In flow cytometry figure " Figure 4 " the percentage of different positive cells for each antibody should be mentioned.
Response: Thank you for the comment. We used representative histograms to show expression patterns of macrophage marker molecules in purified uterine macrophages: many histograms consisted of a single fraction. The expression patterns clearly indicate that the uterine macrophages were polarized to the M2 phenotype. When we examine expression levels of some marker molecules in uterine macrophages treated with and without certain reagents such as ones to induce M1/M2 polarization by flow cytometry, we must compare percentages of positive/negative cells and/or the mean fluorescence intensities of positive cells between the two. We think that the rate of positive cells for each antibody is unnecessary in our present data.
Comment 6: In the discussion section line, 538 references should be cited.
Response: Thank you for the comment. We replaced “previous studies” with “those previous studies” to identify the cited studies (reference #40 and #25). This is now in line 479 on page 13.

Reviewer 2 Report
1. Page 1, line 24-25
Why this part has no reference?
2. Page 1, line 27
Why this part has middle-sentence reference?
3. Page 1, line 29
Why this sentence has multiple reference?
4. Page 1, line 29-36
This part contains no references. Why?
5. Page 1, line 38
muliple reference should be reconsidered in this part
6. Multiple references in page 1, line 40-41
should be reformed.
7. Page 1, line 41
Please define the abbreviation "BMMs" here.
8. All over the part "1. Introduction" there are some multiple and middle-sentence references. Please reform all of them.
9. Page 2, line 49
The abbreviation "CSF1" need to be defined and cleared.
10. Page 2, line 55-63
Why this part has no reference(s)?
Please add proper reference at the end of each sentence of this part.
11. Page 2, line 64
This line has disrupted the continuity of the main text because this line has not logical
connection with its prior paragraph. Please make this connection in order to keep consistency of the text.
12. Page 2, line 70-84
Why this part has no proper reference(s)?
13. Page 2, line 70-84
Why the authors have written this part here? It seems that this part belongs to the part "Discussion"
Please explain the reason of this part
14. Page 6, line 259-261
This part does not belong here (it belongs to the part "materials and methods")
Please reconsider this part.
15. Page 6, line 263
Where is Figure 1??
Note: please insert each figure or table in the nearest place that you mention its name for
the first time (exert this note for all parts of the manuscript)
16. Page 6, line 270-273
Why the authors have written this part here?
17. Page 6, line 273
Why you have inserted reference(s) in the part "Results"?
Please ommit reference(s) from different sections of the part "Results"
Note: in the part "Results" the authors have to speak about only "Results", they should not insert or mention any references, comparisons with the results of other studies, extra explaination about tests and any information that belongs to the part materials and methods
18. Page 6, line 274-275
Why the title is so long here? Could you please reform the title?
19. Page 6, line 276-278
Why the authors have written this part here?
This part is not related to the part "Results"
Please refer to note in comment 17
20. Page 6, line 283
Please insert Figure 2 in the nearest place that you mention its name for the first time
21. Please reorganize all of your figures based on the note in comment 15
22. Please reorganize Figure 3
The words and font color should be reconsidered and become more obvious.
23. About Figure 2
Please recorganize this figure and make its words and their font color more obvious. (The words and information in the figure are not obviously visible)
24. About Figure 3
24.1)
The chart in the right side of figure that its vertical pivot shows Phagocytic cell (%), the horizental pivot of this chart demostrate what kind of information?
24.2)
Why this figure has such a long subtitle with detailed information (line 334-344)?
25. Page 9, line 359-360
Why this title is so long and detailed?
25. Page 9, line 361-364
Please refer to note in comment 17 and ommit this part, it does not belong here.
26. Page 10, line 371-381
Why the authors have inserted reference(s) here?
Why the authors have mentioned this part here? This part belongs to the part "Discussion" not here.
Please recosider this part and refer to note in comment 17 (all of this part should be reformed based on mentioned note)
27. Page 10, line 385-387
Why the title of this figure os so long and detailed? Please reconsider it
28. About figure 4
This figure and other figures in the manuscript must be reconsidered based on
the note in comment 15
29. Page 10, line 388-396
Why the subtitle of this figure contains a lot of detailed data? Is it not better to reconsider this part and transfer all these extra and detailed data to their proper place in the main text?
30. Page 11, line 398-403
Please ommit this part and refer to the note in comment 17
31. Page 11, line 405
Where is Figure 5? Please add Figure 5 to its proper place (in order to do that, please refer to the note in comment 1
32. Page 11, line 407
Why you have inserted reference here?
(Please refer to note in comment 17)
33. Page 11, line 405-412 and line 420-424
This part should be ommited because it does not belong here (refer to comment 17)
34. Page 11, 431-438 and line 439-443
This part should be deleted from here (refer to comment 17)
35. Page 12, line 450-451
Please ommit this part. It does not belong here
36. About all of the figures in the manuscript:
First: the titles of some of the figure are so long and contain alot of detailed data that belings to the main text( not subtitle of figure). Thus, please reconsider the subtitle of the figures
Second: all the information in the figures
must be obbiously visible. Thus, please reconsider the font size, font color and other necessary changes in the figures and make them clearly visible
Third: each figure shoud be in the nearest place that the authors mentioned its name for the first time
37. About all sections of the part "Results"
First: Please reconsider all middle-sentence and multiple references in this part
Second: please only tell "Results" in this part. Any extra and irrelevant explainations should be ommitted. (Refer to comment 17)
38. About the part "Discussion"
Reform all multiple and middle-sentence references
39. Please check and adjust the "Reference list" based on the regulations of reference list of journal. (Titles, doi, the name of journal and ... )
Author Response
Responses to the comments of Reviewer 2
We have revised the manuscript to address the comments of Reviewer 2 as outlined below. We have also highlighted the changes within the document in blue.
Comment 1: Page 1, line 24-25. Why this part has no reference?
Response: According to the comment, we added a reference (#1). This is now in line 25 on page 2.
Comment 2: Page 1, line 27. Why this part has middle-sentence reference?
Response: Thank you for the reviewer’s comment. We removed this reference because references #2 and #3 cover the evidence of the whole sentence.
Comment 3: Page 1, line 29. Why this sentence has multiple reference?
Response: Thank you for the reviewer’s comment. There have been published so many articles in which multiple references are cited in a sentence. Such articles are usually seen in Biomedicines. Reference #2 and #3 are representative and valuable articles suitable for citation in the sentence. Thus, we cited both articles.
Comment 4: Page 1, line 29-36. This part contains no references. Why?
Response: According to the comment, we added references in the first sentence (references #2 and #3) and the second sentence (reference #1) of the part. This is now in lines 32 and 34 on page 2.
Comment 5: Page 1, line 38. muliple reference should be reconsidered in this part
Response: Thank you for the reviewer’s comment. There have been published so many papers in which multiple references are cited in one sentence. Such articles are usually seen in Biomedicines. We removed one reference. The other two (references #4 and #5) are valuable articles suitable for citation in the sentence.
Comment 6: Multiple references in page 1, line 40-41. should be reformed.
Response: Thank you for the reviewer’s comment. Again, there have been published so many papers in which multiple references are cited in one sentence. Such articles are usually seen in Biomedicines. References #7 and #8 as well as #4 and #9 are valuable articles suitable for citation in each sentence. Thus, we did not change the citations in the revised manuscript.
Comment 7: Page 1, line 41. Please define the abbreviation "BMMs" here.
Response: In the previous sentence (line 27), the abbreviation "BMMs" is already defined.
Comment 8: All over the part "1. Introduction" there are some multiple and middle-sentence references. Please reform all of them.
Response: Thank you for the reviewer’s comment. There have been published so many papers in which multiple references and middle-sentence references are cited in one sentence. Such articles are usually seen in Biomedicines. We would like to avoid multiple references as much as possible. However, in many cases, we could not select one reference because selected references are equally valuable articles suitable for citation in every sentence. We avoided middle-sentence references as much as possible. However, we used middle-sentence references in two places ( reference #22, #23, and #24; reference #26 and #27) of the introduction section due to the clear identification of corresponding objects (cells).
Comment 9: Page 2, line 49. The abbreviation "CSF1" need to be defined and cleared.
Response: In the previous sentence (line 30), the abbreviation "CSF1" is already defined.
Comment 10: Page 2, line 55-63. Why this part has no reference(s)? Please add proper reference at the end of each sentence of this part.
Response: Thank you for the reviewer’s comment. We added two references (#4 and #9) in the first sentence of this part. We described our point of view in the following sentences, and thus there are no references in the following.
Comment 11: Page 2, line 64. This line has disrupted the continuity of the main text because this line has not logical connection with its prior paragraph. Please make this connection in order to keep consistency of the text.
Response: We appreciate the reviewer’s comment. We moved the first two sentences to the middle part (lines 76-79 in the revised manuscript) of this paragraph to make a logical connection.
Comment 12: Page 2, line 70-84. Why this part has no proper reference(s)?
Response: As we described “ In these mixed cultures,--- (first sentence of this part; line 71 in the revised manuscript)”, this part consists of descriptions explaining tissue-resident Mø propagation by mixed culture in references #22-24 (lines 69 and 70). Thus, we think references are unnecessary in this part.
Comment 13: Page 2, line 70-84. Why the authors have written this part here? It seems that this part belongs to the part "Discussion". Please explain the reason of this part
Response: Thank you for the reviewer’s comment. We rewrote the sentences to fit the part “Introduction” mainly by removing the last sentence in the paragraph. This is now in lines 71-85 on page 2.
Comment 14: Page 6, line 259-261. This part does not belong here (it belongs to the part "materials and methods") Please reconsider this part.
Response: We wrote this sentence as a preface in order to easily understand what we examined in the subsection of the “Results”. This kind of description is frequently seen in the “Results” section in many articles and such articles are also seen in Biomedicine. Thus, we did not remove this part but rewrote the sentence. This is now in lines 274-276 on page 6 of the revised manuscript.
Comment 15: Page 6, line 263. Where is Figure 1??
Note: please insert each figure or table in the nearest place that you mention its name for the first time (exert this note for all parts of the manuscript)
Response: Thank you for the reviewer’s comment. Figure 1 is vertically long and fits into a space of almost one page. We tried to place Figures as close as possible following the sentences of figure description first seen in the text. However, Figure 1 is still on page 7 in the revised manuscript.
Comment 16: Page 6, line 270-273. Why the authors have written this part here?
Response: Thank you for the reviewer’s comment. We think this part is necessary clearly to understand the following result subsection. We rewrote this part by removing the latter clause with references. This is now lines 284-286 on page 6.
Comment 17: Page 6, line 273. Why you have inserted reference(s) in the part "Results"? Please ommit reference(s) from different sections of the part "Results"
Note: in the part "Results" the authors have to speak about only "Results", they should not insert or mention any references, comparisons with the results of other studies, extra explanation about tests and any information that belongs to the part materials and methods
Response: Thank you for the reviewer’s comment. As described in the response to comment 16, we removed the sentence (the latter clause) with the references.
Comment 18: Page 6, line 274-275. Why the title is so long here? Could you please reform the title?
Response: Thank you for the reviewer’s comment. We rewrote the subtitle concisely. This is now in line 287 on page 6 of the revised manuscript.
Comment 19: Page 6, line 276-278. Why the authors have written this part here? This part is not related to the part "Results". Please refer to note in comment 17.
Response: We wrote this sentence as a summary in this result subsection. This kind of description method is frequently seen in the “Results” section in many articles and such articles are also seen in Biomedicine. I think this first sentence is necessary for connecting the following sentence. Thus, we did not remove the whole sentence but rewrote it. This is now in lines 288-289 on page 6 of the revised manuscript.
Comment 20: Page 6, line 283. Please insert Figure 2 in the nearest place that you mention its name for the first time.
Response: Thank you for the reviewer’s comment. Figure 2 is also vertically long and fits into a space of almost one page. We tried to place Figures as close as possible following the sentences of figure description first seen in the text. However, Figure 2 is still on page 8 of the revised manuscript.
Comment 21: Please reorganize all of your figures based on the note in comment 15.
Response: Thank you for the reviewer’s comment. We definitely understand suitable positions of Figures in the text. We tried to place Figures as close as possible following the sentences of figure description first seen in the text. We think that the journal editors also face trouble when they try to replace Figures in the text of our manuscript.
Comment 22: Please reorganize Figure 3. The words and font color should be reconsidered and become more obvious.
Response: Thank you for the reviewer’s comment. According to the comment, we used a larger font size as well as a bold font in some places in Figure 3 in the revised manuscript.
Comment 23: About Figure 2. Please recorganize this figure and make its words and their font color more obvious. (The words and information in the figure are not obviously visible)
Response: Thank you for the reviewer’s comment. According to the comment, we used a larger font size as well as a bold font in Figure 2 in the revised manuscript.
Comment 24.1): About Figure 3. The chart in the right side of figure that its vertical pivot shows Phagocytic cell (%), the horizental pivot of this chart demostrate what kind of information?
Response: We appreciate the reviewer’s comment. We added “cells adherent to the Petri dish” as the item of the horizontal axis in Figure 3E.
Comment 24.2): About Figure 3. Why this figure has such a long subtitle with detailed information (line 334-344)?
Response: Thank you for the reviewer’s comment. Referring lines 334-344, we think that the comment noted the caption of Figure 2. We rewrote the figure caption to be concise. This is now in lines 316-320 on page 8 of the revised manuscript.
Comment 25: Page 9, line 359-360. Why this title is so long and detailed?
Response: According to the comment, we rewrote the subtitle to be concise. This is now in lines 365-366 on page 10 of the revised manuscript.
Comment 26: Page 9, line 361-364. Please refer to note in comment 17 and ommit this part, it does not belong here.
Response: Thank you for the reviewer’s comment. We wrote this sentence as a preface in order to easily understand what we examined in the subsection of the “Results”. This kind of description is frequently seen in the “Results” section in many articles, and such articles are also seen in Biomedicine. Thus, we did not remove this part but rewrote the sentence concisely. This is now in lines 367-368 on page 10 of the revised manuscript.
Comment 27: Page 10, line 371-381. Why the authors have inserted reference(s) here?
Why the authors have mentioned this part here? This part belongs to the part "Discussion" not here.
Please recosider this part and refer to note in comment 17 (all of this part should be reformed based on mentioned note)
Response: According to the comment, we removed the sentences with references and rewrote a sentence. This is now in lines 375-378 on page 10 in the revised manuscript.
Comment 28: Page 10, line 385-387. Why the title of this figure os so long and detailed? Please reconsider it.
Response: According to the comment, we rewrote the title to be concise. This is now in line 381 on page 10 of the revised manuscript.
Comment 29: About figure 4. This figure and other figures in the manuscript must be reconsidered based on the note in comment 15.
Response: According to the comment, we could put Figure 4 just below the description of the text related to the figure. This is now on page 10 of the revised manuscript.
Comment 30: Page 10, line 388-396. Why the subtitle of this figure contains a lot of detailed data? Is it not better to reconsider this part and transfer all these extra and detailed data to their proper place in the main text?
Response: According to the comment, we rewrote the figure caption to be concise. This is now in lines 382-386 on page 10 of the revised manuscript.
Comment 31: Page 11, line 398-403. Please ommit this part and refer to the note in comment 17.
Response: Thank you for the reviewer’s comment. We wrote the sentences as a preface in order to easily understand what we examined in the subsection of the “Results”. This kind of description is frequently seen in the “Results” section in many articles and such articles are also seen in Biomedicine. Thus, we did not remove this part but rewrote the sentences concisely. This is now in lines 388-391 on page 10 of the revised manuscript.
Comment 32: Page 11, line 405. Where is Figure 5? Please add Figure 5 to its proper place (in order to do that, please refer to the note in comment 1.
Response: According to the comment, we could put Figure 5 just below the description of the text related to the figure. This is now on page 11 of the revised manuscript.
Comment 33: Page 11, line 407. Why you have inserted reference here? (Please refer to note in comment 17)
Response: According to the comment, we remove the reference and the related phrase.
Comment 34: Page 11, line 405-412 and line 420-424. This part should be ommited because it does not belong here (refer to comment 17)
Response: According to the comment, we remove the sentences with references.
Comment 35: Page 11, 431-438 and line 439-443. This part should be deleted from here (refer to comment 17).
Response: According to the comment, we remove the sentences with references and rewrote these parts to be one sentence in which we described the reason why we examined the expression in this subtitle part of the Results. This is now in lines 424-426 on page 12 in the revised manuscript.
Comment 36: Page 12, line 450-451. Please ommit this part. It does not belong here.
Response: According to the comment, we remove the sentences with references.
Comment 37: About all of the figures in the manuscript:
First: the titles of some of the figure are so long and contain alot of detailed data that belings to the main text( not subtitle of figure). Thus, please reconsider the subtitle of the figures. Second: all the information in the figures must be obbiously visible. Thus, please reconsider the font size, font color and other necessary changes in the figures and make them clearly visible. Third: each figure shoud be in the nearest place that the authors mentioned its name for the first time
Response: Thank you for the reviewer’s comments. According to the first comment, we rewrote the titles and subtitles of the figure captions to be concise as much as possible. According to the second comment, we used a larger font size as well as a bold font in the Figures in the revised manuscript. Again, we definitely understand suitable positions of Figures in the text. We tried to place Figures as close as possible following the sentences of figure description first seen in the text. We think that the journal editors also face trouble when they try to replace Figures in the text of our manuscript.
Comment 38: About all sections of the part "Results"
First: Please reconsider all middle-sentence and multiple references in this part. Second: please only tell "Results" in this part. Any extra and irrelevant explainations should be ommitted. (Refer to comment 17)
Response: Thank you for the reviewer’s comments. According to the first comment, we removed sentences with references (we already mentioned this several times above). At the beginning of subsections of the “Results” section, we usually wrote a sentence as a preface in order to easily understand what we examined in the subsection of the “Results”. This kind of description is frequently seen in the “Results” section in many articles and such articles are also seen in Biomedicine. Thus, we did not remove this kind of sentence but rewrote the sentence to be concise (we already mentioned this several times above).
Comment 39: About the part "Discussion". Reform all multiple and middle-sentence references.
Response: Thank you for the reviewer’s comment. Again, there have been published so many papers in which multiple references and middle-sentence references are cited in one sentence. Such articles are usually seen in Biomedicines. We would like to avoid multiple references as much as possible in the revised manuscript. However, in many cases, we could not select one reference because selected references are valuable articles suitable for citation in every sentence. We avoided middle-sentence references as much as possible. However, we used middle-sentence references in the discussion section due to the clear identification of corresponding objects.
Comment 40: Please check and adjust the "Reference list" based on the regulations of reference list of journal. (Titles, doi, the name of journal and ... )
Response: Thank you for the reviewer’s comment. We checked the “Reference list”.

Round 2
Reviewer 2 Report
I dont have more comment. Thank you for considering my suggestions